# Cycling as the Best Sub-8-Hour Performance Predictor in Full Distance Triathlon

**DOI:** 10.3390/sports7010024

**Published:** 2019-01-18

**Authors:** Caio Victor Sousa, Lucas Pinheiro Barbosa, Marcelo Magalhães Sales, Patrick Anderson Santos, Eduard Tiozzo, Herbert Gustavo Simões, Pantelis Theodoros Nikolaidis, Beat Knechtle

**Affiliations:** 1Graduate Program in Physical Education, Catholic University of Brasília, Brasília 71966-700, DF, Brazil; cvsousa89@gmail.com (C.V.S.); lduarte.barbosa@gmail.com (L.P.B.); patricksantospas@gmail.com (P.A.S.); hgsimoes@gmail.com (H.G.S.); 2Miller School of Medicine, University of Miami, Miami, FL 33136, USA; etiozzo@med.miami.edu; 3Department of Physical Education, Goiás State University, Quirinopolis 75860-000, GO, Brazil; marcelomagalhaessales@gmail.com; 4Exercise Physiology Laboratory, Nikaia 12244, Greece; pademil@hotmail.com; 5Institute of Primary Care, University of Zurich, CH-8006 Zurich, Switzerland

**Keywords:** swimming, running, exercise training, athlete

## Abstract

For any triathlon distance (short, Olympic, half-distance and full-distance), competitors spend more time cycling than swimming or running, but running has emerged as the discipline with the greatest influence on overall performance at the Olympic distance. However, there is a lack of evidence on which discipline has the greatest influence on performance in the overall full-distance triathlon (3.8 km swimming/180 km cycling/42.195 km running), especially for the fastest performing athletes of all time. The total race times of 51 fastest triathletes (sub-8-hour) were studied, while for the split times, a sample of 44 participants was considered. The discipline that seemed to better predict total race time was cycling (coefficient = 0.828; *p* < 0.001), followed by running (coefficient = 0.726; *p* < 0.001) and swimming (coefficient = 0.476; *p* < 0.001). Furthermore, cycling was the discipline with the highest performance improvement over the years, whereas running had a slightly decrease. In conclusion, cycling seems to be the discipline with greater influence in final result for the full-distance triathlon.

## 1. Introduction

In the very beginning of triathlon in 1978, a competition was held in Hawaii in which 12 athletes completed 2.4 miles (3.8 km) of swimming, 112 miles (180 km) of cycling and followed by 26.2 miles (42.195 km) of running, with the best race time of 11 h, 46 min and 58 s [1]. This official full-distance triathlon has been growing in popularity, with new race routes being added every year worldwide and elite athletes reaching great performances in each race [1,2]. For instance, nowadays, the performance of 11:56:58 is not even eligible to qualify an amateur athlete below 55 years old to the Ironman World Championship [2].

The analysis of triathlon performance has been useful and necessary for both amateurs and professionals [1,3,4]. Thus, there are studies investigating several aspects such as physiological, nutritional and strategies of pace in every modality for a better performance in a full-distance triathlon [5,6,7,8].

In that regard, new scientific evidence about triathlon has come up every year in order to better serve the athletes and coaches, including trending and performance analysis. For instance, although most time of the race is spent on cycling in any triathlon distances (short, Olympic, half-distance and full-distance) [2,8], there are studies reporting that running is the discipline that most influences the overall performance (race time) in the Olympic distance (1.5 km swim; 40 km cycle; 10 km run) [9,10,11].

The Olympic distance triathlon has several differences in comparison to full-distance triathlon, starting with the profile of elite athletes. The top 20 athletes competing at Olympic distances are, in average, 10 years younger than the best 20 athletes competing in full-distance triathlon [11,12,13]. Physiological parameters also differ, since Olympic distance has a much greater glycolytic activity than full distance [14]. Finally, drafting rules and formation of cycling packs, which reduce wind resistance to conserve energy, are allowed only in the Olympic distance [3].

There are some analyses of the full-distance triathlon involving physiological and morphological parameters to predict overall performance [12]. However, there is a lack of evidence regarding which modality most influences the overall performance in the full-distance triathlon, especially among the fastest elite triathletes. Moreover, all performance trends published so far were focused on age groups (amateurs), which may have a completely different strategy than professionals. The analysis of sub-8-hour performance enables the selection of the top-level athletes. Thus, the aim of present study was to identify which modality can better predict the total race time and explain performance trends in each discipline in a sub-8-hour full-distance triathlon from 1997 to 2018.

## 2. Materials and Methods

All procedures used in the study were approved by the Institutional Review Board of Kanton St. Gallen, Switzerland, with a waiver of the requirement for informed consent of the participants given the fact that the study involved the analysis of publicly available data.

We obtained data from publicly available databases (www.ironman.com, www.challenge-family.com and www.challenge-roth.com). All official total race times among Ironman and Challenge races, including World and Continental Championships, performed under 08:00:00 from 1997 up to November 2018 were recorded. Forty-nine records were initially eligible to be included, but three were excluded because split times were not available, and six more were excluded because transition times were not separated from swim/cycle/run times. Therefore, the total sample included in the analysis was composed by 51 races, but split analyses included 44 only.

Initially, an exploratory analysis of the data was performed, with median, 25 and 75 percentiles, as well as the fastest and slowest sub-8-hour performances displayed in Table 1. Further, all data were transformed in seconds and then a stepwise multiple linear regression was performed using total race time as the dependent variable and splits and transitions times as independent variables. A bivariate association analysis was also performed between total race time and each split time, the Spearman correlation coefficient was applied. A linear regression from each discipline (swimming/cycling/running) with the year of the race was also performed. The significance level was set as 5% (*p* < 0.05), and all procedures were performed using SPSS v21.0 (IBM SPSS Statistics for Windows, Version 21.0. 2012. Armonk, NY, USA: IBM Corp).

## 3. Results

The records included data from 14 different races, the Challenge Roth (Germany) being the race with more sub-8-hour performances, including the World Record in performance (07:35:41) by the Olympic Gold Medalist (2008) and two times Ironman World Champion (2015 and 2016), Jan Frodeno from Germany. The discipline with the greatest proportion of the total race time was cycling (54.13% ± 1.06), followed by running (34.83% ± 1.02), swimming (10.20% ± 0.64) and the sum of transition times (0.84% ± 0.18).

The best model from the stepwise multiple regression included Swimming, Cycling and Running split times with adjusted (_aj_) coefficient of determination of 67.7% (R^2^ = 0.835; R^2^_aj_ = 0.677; *p* < 0.001). The discipline that best predicts total race time was cycling (coefficient = 0.868; *p* < 0.001), followed by running (coefficient = 0.726; *p* < 0.001) and swimming (coefficient = 0.476; *p* < 0.001). Further, the correlation analyses indicate that the total race time was best associated with Cycling performance (*r* = 0.520), followed by Swimming (*r* = 0.327) and Running (*r* = 0.151). See Figure 1 for details with the standardized beta from multiple regression and Spearman coefficient.

The linear and non-linear regressions of total performances and race year did not reach a significant result and the generation of the equation was not reliable (R^2^ < 0.5) to predict a sub 7h30min performance (Figure 2).

The linear regression shows a negative and significant slope for cycling from 1999 to 2018, whereas it shows a significant and positive slope for running. Swimming did not show a significant slope (Figure 3). The results suggest that performance has improved in cycling, remained the same in swimming and decreased in the running throughout the years.

## 4. Discussion

The main finding of the present study was that sub-8-hour performances in full-distance triathlon are better predicted by cycling performance, as compared to swimming or running. Furthermore, the splits for cycling have improved over the years whereas the opposite was observed for running.

It is not novel that cycling has a great time contribution in the overall time in all triathlon distances [2], including the top full triathlon performances. However, the impact of each discipline for the overall race time seems to differ among various triathlon distances. Peeling and Landers [15] previously reported that the swimming in shorter triathlon races (i.e., short and Olympic) will determine if the athlete will cycle in the leading or chasing pack. Therefore, riding within a pack would turn the cycling discipline less competitive and more about ‘saving energy’ for the running leg, making cycling data very homogeneous in elite athletes and reducing the general influence of it in the overall race time.

In that regard, there is evidence showing that running has the greatest association with overall Olympic distance performance [9,10]. Furthermore, recently Ofoghi, Zeleznikow, Macmahon, Rehula and Dwyer [11] performed a more sophisticated analysis and concluded that swimming and running are more important for the general success than cycling in an Olympic triathlon, even stating that cycling has little or no influence in the outcome.

However, in a full distance triathlon it is not allowed to draft during cycling, thus there are no packs. Moreover, since swimming has the smallest time contribution in a full-distance it is absolutely possible that a poor swimmer make a great overall race time with good cycling and running performances, reducing the influence of swimming, as our results indicate. Therefore, an explanation for which full-distance triathlons have opposite results in comparison to previous studies with Olympic distance triathlon is due to the regulations of each race. Short and Olympic triathlons allow drafting in cycling, whereas full-distance does not, therefore, since cycling is the discipline that comprehends the largest portion of time in all distances (short, Olympic and full-distance) it is reasonable to infer that it would be more important if the athletes had to do it solo.

The linear regression of split times showed that cycling times have improved since the first sub-8-hour performance. There are several aspects regarding cycling performance that have improved in the last two decades, including the methods of training and race strategies, but we believe that aerodynamics is the most responsible for performance improvement on cycling. Ever since the advanced methods to test a bike ride aerodynamics [16], such as wind tunnels, a lot has changed in the bike frame to adapt the athlete to a better aero position and less energy expenditure [17,18], including the geometry of the frame itself, aero wheels, aero helmets, aero outfit, all to reduce the drag and make the athlete faster and more comfortable [17,18]. Unsurprisingly, cycling records are beaten every year in full distance triathlon races. On the other hand, as a result of a faster pace on cycling, the subsequent running performances have decreased from 1999 to 2018, which could indicate a change in strategy for a better final performance in a full-distance event. 

The implications of such results are in accordance with current literature regarding the physiological and biomechanical implications in triathlon. There are evidences that cycling has a stronger influence of peripheral fatigue of active muscles than running [19], whereas for running the mechanical efficiency in regard to the cardiometabolic variables is more important [20]. Since, throughout the years, athletes racing sub-8-hour full distance triathlon are becoming faster in cycling and slightly slower in running, it is reasonable to infer that a greater neuromuscular fatigue is elicited as a result of pushing harder in cycling, which is closely related with a decreased efficiency and impaired performance in a subsequent running [20,21].

The results are applicable especially for elite athletes racing full distance triathlon events to determine training focus and race strategy. For non-elite athletes these results should be interpreted with caution, since the data only included the fastest performances ever. A possible limitation of the present analysis is that the inclusion of only sub-8-hour performances excluded races locations with tough environmental conditions (temperature and course), and that the race Challenge Roth is the one with most sub-8-hour performances. However, we highlight that this is the first performance analysis of the top-level athletes in full distance triathlon, and it could be beneficial for athletes and coaches to better define their training and racing strategies.

## 5. Conclusions

In conclusion, cycling seems to be the discipline with greater influence in the overall result for the full-distance triathlon. Cycling performance has improved among the sub-8-hour full Ironman finishers, possibly due to improvement in aerodynamics and/or racing strategy. Future research should include a bigger sample of athletes, including those with performances exceeding 8 h and amateur athletes.

## Figures and Tables

**Figure 1 sports-07-00024-f001:**
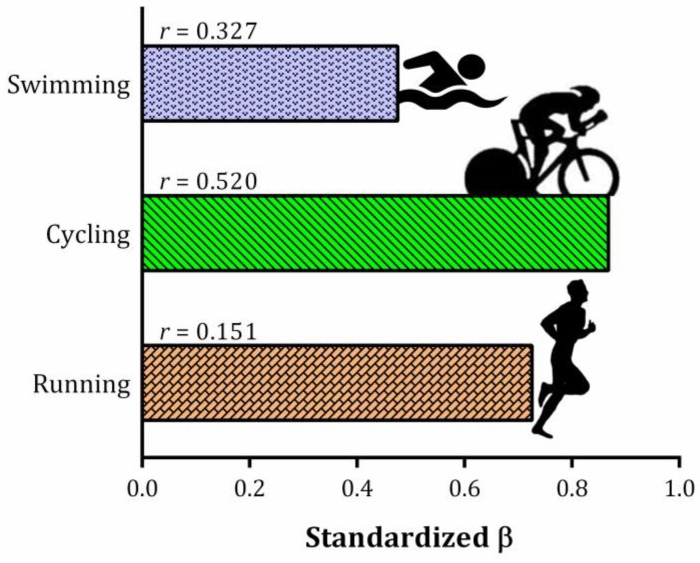
Standardized coefficient from stepwise multiple regression using total race time as dependent variable of sub-8-hour performance in full distance triathlon (3.8 km swimming/180 km cycling/42.195 km running) and Spearman correlation coefficient between each variable and overall race time.

**Figure 2 sports-07-00024-f002:**
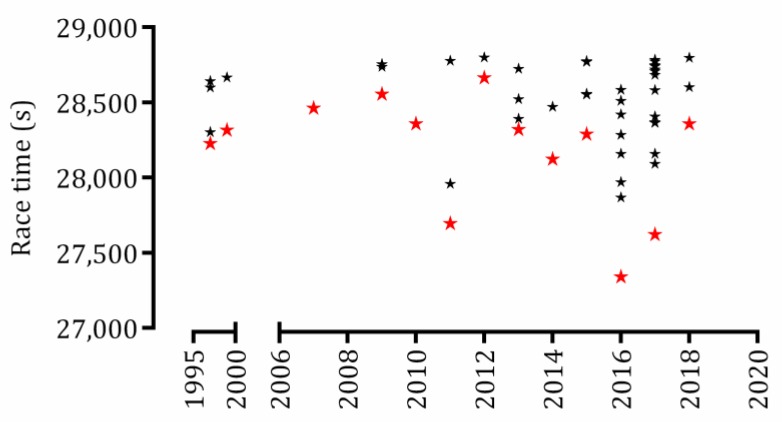
Dispersion of sub-8-hour performances in full distance triathlon from 1997 to 2018. Red stars indicate the best of each year.

**Figure 3 sports-07-00024-f003:**
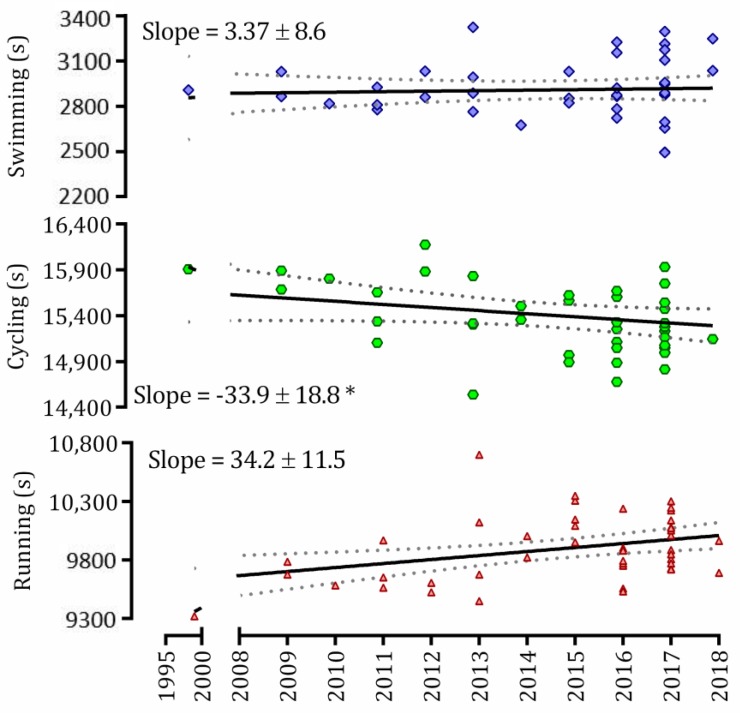
Dispersion and linear regression of split-times (seconds) of sub-8-hour performances in full distance triathlon from 1999 to 2018; *: *p* < 0.05.

**Table 1 sports-07-00024-t001:** Total race, splits and transition times of sub-8-hour full distance triathlon (h:min:s).

Splits	Median (25–75 Percentile)	Min	Max
Overall	07:55:12 (07:51:30–07:58:29)	07:35:41	07:59:59
Swim	00:48:01 (00:46:43–00:49:47)	00:41:33	00:55:23
Cycle	04:15:47 (04:11:51–04:21:41)	04:02:17	04:29:34
Run	02:44:25 (02:41:17–02:48:03)	02:35:21	02:58:18
T1	00:02:22 (00:01:58–00:02:35)	00:01:19	00:03:19
T2	00:01:39 (00:01:15–00:02:06)	00:00:54	00:02:47

T1—transition 1 (swim to cycle); T2—transition.

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
