# Peer review of "Cycling as the Best Sub-8-Hour Performance Predictor in Full Distance Triathlon"

_sports, 2019, doi:10.3390/sports7010024_

Reviewer 1 Report

The paper is quite interesting and significant for sport science as well as for sports training. The Introduction is quite short and the research objectives should be better justified. The authors should look into some more recent refernces. The Journal of Human Kinetics has published several papers on the triathlon. The methods section seems fine, while in the results section please use SI units and apropriate abreviations in Figures 3 & 4. It should be s not sec or seconds. In the discussion the authors should try to explain the results in more detail, rather than just comparing them to those of other researchers. Perhaps the metabolic consequences of swimming, running and cycling should be considered. Last but most important is the correction of English, style and grammar. Please consult a native speaker to improve the English in the text.

Author Response

Reviewer #1:

The paper is quite interesting and significant for sport science as well as for sports training.

COMMENT #1: The Introduction is quite short and the research objectives should be better justified. The authors should look into some more recent references. The Journal of Human Kinetics has published several papers on the triathlon.

REPLY: We thank the expert reviewer for their comments. However, we disagree with the reviewer regarding the size of the introduction, we believe it is well-fitted according to the scope of the manuscript. Within the last two years (2017-2018) the Journal of Human Kinetics only published one manuscript on triathlon (Marcos-Serrano, Olcina, Crespo, Brooks, & Timon, 2018) which has already been referenced in the manuscript. We further referenced another manuscript ‘ahead of print’ from this same journal (Sousa et al., 2019).

COMMENT #2: The methods section seems fine, while in the results section please use SI units and appropriate abbreviations in Figures 3 & 4. It should be s not sec or seconds.

REPLY: We agree with the reviewer and changed the figures to the appropriate abbreviations in Figures 3 and 4.

COMMENT #3: In the discussion the authors should try to explain the results in more detail, rather than just comparing them to those of other researchers. Perhaps the metabolic consequences of swimming, running and cycling should be considered.

REPLY: We thank the reviewer for the comment and we also agree that the metabolic consequences of each discipline should be more explored to explain the performance in subsequent disciplines. However, regarding the results of our manuscript, literature seems to point that it is a matter of strategy and regulations of the race.

We further, highlight that scientific evidences regarding the effects of physiological changes during a discipline (swim, cycle) affects the other disciplines (cycle, run) in triathlon is very poor, especially in full triathlon distance. Nevertheless, we discuss the possible mechanisms the may have influenced the results. See lines 162 to 169 highlighted in red.

COMMENT #4: Last but most important is the correction of English, style and grammar. Please consult a native speaker to improve the English in the text.

REPLY: We thank the reviewer for the comment. An English native speaker has reviewed the manuscript.

Reviewer 2 Report

The research presented in this paper focuses on the analysis of the impact that each of the different disciplines of which a triathlon consists on the final result of the race. The study concludes that for races with a duration of less than 8 hours, it is cycling that best predicts the final result.

The fastest records (less than 8 hours) of an acceptable number of races have been taken into account to carry out the study. The article is presented clearly and only a few typographical errors have been detected, which are pointed out at the end of this report.

The paper also reviews the relevant literature and uses statistical tools consistent with the type of study to be performed. It would have been desirable to make a division of the sample in the study to discard the effect caused by the place of the race, although this reviewer thinks that it would not have effect on the main conclusions of the paper.

Minor edits:

Line 99 "Spearmen" by "Spearman"                

Author Response

Reviewer #2:

The research presented in this paper focuses on the analysis of the impact that each of the different disciplines of which a triathlon consists on the final result of the race. The study concludes that for races with a duration of less than 8 hours, it is cycling that best predicts the final result.

The fastest records (less than 8 hours) of an acceptable number of races have been taken into account to carry out the study. The article is presented clearly and only a few typographical errors have been detected, which are pointed out at the end of this report.

The paper also reviews the relevant literature and uses statistical tools consistent with the type of study to be performed. It would have been desirable to make a division of the sample in the study to discard the effect caused by the place of the race, although this reviewer thinks that it would not have effect on the main conclusions of the paper.

Minor edits:

COMMENT #1: Line 99 "Spearmen" by "Spearman"

REPLY: We thank the reviewer for pointing this error. It has been corrected.

Marcos-Serrano, M., Olcina, G., Crespo, C., Brooks, D., & Timon, R. (2018). Urinary Steroid Profile in Ironman Triathletes. J Hum Kinet, 61, 109-117. doi:10.1515/hukin-2017-0130

Sousa, C., Aguiar, S., Olher, R., Sales, M., Moraes, M., Nikolaidis, P., . . . Simões, H. (2019). Hydration status after an Ironman triathlon: a meta-analysis. Journal of human kinetics, Ahead of print. doi:10.2478/hukin-2018-0096